# Functional Impact of Neuro-Vascular Bundle Preservation in High Risk Prostate Cancer without Compromising Oncological Outcomes: A Propensity-Modelled Analysis

**DOI:** 10.3390/cancers15245839

**Published:** 2023-12-14

**Authors:** Marc A. Furrer, Niranjan Sathianathen, Brigitta Gahl, Patrick Y. Wuethrich, Gianluca Giannarini, Niall M. Corcoran, George N. Thalmann

**Affiliations:** 1Department of Urology, Inselspital, Bern University Hospital, University of Bern, 3012 Bern, Switzerland; george.thalmann@insel.ch; 2Department of Urology, Solothurner Spitäler AG, Kantonsspital Olten, 4600 Olten, Switzerland; 3Bürgerspital Solothurn, 4500 Solothurn, Switzerland; 4Department of Urology, The University of Melbourne, Royal Melbourne Hospital, Parkville, VIC 3050, Australia; niranjan.sathianathen@wh.org.au (N.S.); niall.corcoran@wh.org.au (N.M.C.); 5Department of Anaesthesiology and Pain Medicine, Inselspital, Bern University Hospital, University of Bern, 3010 Bern, Switzerland; patrick.wuethrich@insel.ch; 6Clinical Trials Unit Bern, University of Bern, 3010 Bern, Switzerland; brigitta.gahl@usb.ch; 7Urology Unit, Santa Maria della Misericordia University Hospital, 33100 Udine, Italy; gianluca.giannarini@asufc.sanita.fvg.it; 8Department of Urology, Western Health, St. Albans, VIC 3021, Australia

**Keywords:** open radical prostatectomy, nerve sparing, erectile function recovery, urinary continence, tumor recurrence

## Abstract

**Simple Summary:**

In this study, we investigated the functional and oncological outcomes of low and intermediate risk compared to high risk prostate cancer patients undergoing radical prostatectomy with attempted neurovascular bundle preservation. We found that satisfactory urinary continence and erectile function recovery is possible without compromising oncological outcomes. Therefore, preservation of the neurovascular bundle should be considered in carefully selected patients with high risk disease. Future studies should develop risk stratification tools to identify which high risk prostate cancer cases are suitable for nerve sparing.

**Abstract:**

Nerve sparing (NS) is a surgical technique to optimize functional outcomes of radical prostatectomy (RP). However, it is not recommended in high risk (HR) cases because of the risk of a positive surgical margin that may increase the risk of cancer recurrence. In the last two decades there has been a change of perspective to the effect that in well-selected cases NS could be an oncologically safe option with better functional outcomes. Therefore, we aim to compare the functional outcomes and oncological safety of NS during RP in men with HR disease. A total of 1340 patients were included in this analysis, of which 12% (*n* = 158) underwent non-NSRP and 39% (*n* = 516) and 50% (*n* = 666) uni- and bilateral NSRP, respectively. We calculated a propensity score and used inverse probability of treatment weighting (IPTW) to balance the baseline characteristics of Pca patients undergoing non-NSRP and those having uni- and bilateral NSRP, respectively. NS improved functional outcomes; after IPTW, only 3% of patients having non-NSRP reached complete erectile function recovery (without erectile aid) at 24 months, whereas 22% reached erectile function recovery (with erectile aid), while 87% were continent. Unilateral NS increased the probability of functional recovery in all outcomes (OR 1.1 or 1.2, respectively), bilateral NS slightly more so (OR 1.1 to 1.4). NSRP did not impact the risk of any recurrence (HR 0.99, 95%CI 0.73–1.34, *p* = 0.09), and there was no difference in survival for men who underwent NSRP (HR 0.65, 95%CI 0.39–1.08). There was no difference in cancer-specific survival (0.56, 95%CI 0.29–1.11). Our study found that NSRP significantly improved functional outcomes and can be safely performed in carefully selected patients with HR-PCa without compromising long term oncological outcomes.

## 1. Introduction

Prostate cancer is the most common solid organ malignancy in men in Western countries [1]. The incidence is expected to rise as the population ages and life expectancy increases. At diagnosis, the vast majority of cases is organ-confined [2]. Radical prostatectomy remains the gold standard among radical treatments in eligible patients.

The oncological aim of radical prostatectomy is removal of the entire prostate without leaving cancerous tissue behind. However, achieving excellent functional outcomes—urinary continence and erectile function—are almost equally important to maintain the patient’s quality of life after radical prostatectomy [3]. In a high proportion of patients, urinary incontinence is the most feared complication of radical prostatectomy, as it can significantly impact patients’ quality of life. Continence rates 12 months postoperatively range from 69% to 96% [4]; the significant range can be explained by various definitions of urinary continence and accuracy of different measurement methods. A number of factors have been identified for post-prostatectomy incontinence, including neurovascular bundle preservation, patient characteristics such as diabetes mellitus or vascular disease, surgeon experience, and surgical approach and precision as well as the methods used to collect and report data [5].

As such, meticulous surgery with preservation of the key anatomic structures for urinary continence and potency is considered fundamental to improve functional outcomes and reduce peri- and postoperative complications. Historically, the nerve sparing technique was avoided in high risk prostate cancer patients due to the risk of extraprostatic extension and the subsequent risk of a positive surgical margin that is associated with increased recurrence rates [6]. However, a wider resection that does not preserve the neurovascular bundles compromises functional outcomes [7,8].

In this regard, accurate prediction of extraprostatic extension in prostate cancer is necessary when planning radical prostatectomy. Multiparametric magnetic resonance imaging (mpMRI) of the prostate, which is increasingly being performed prior to targeted prostate biopsy, can be beneficial in the detection of extraprostatic extension [9].

Recently, Nyarangi-Dix et al. developed a prediction model for patient-tailored risk stratification for the presence of extraprostatic disease. They combined clinical parameters (i.e., ISUP grade, clinical T-stage, PSA, core involvement in %, and percentage of positive cores of prostate biopsy) and mpMRI. They concluded that their risk model can accurately predict extraprostatic extension of prostate cancer, and as such may be useful when planning nerve-sparing radical prostatectomy [10].

Several studies have reported favorable oncological outcomes in high risk prostate cancer patients undergoing radical prostatectomy [11,12]. This is in line with our recent study reporting that nerve sparing radical prostatectomy can be attempted without compromising long-term oncological outcomes [13]. Reported ten-year overall and cancer specific mortality-free survival in high risk prostate cancer patients undergoing radical prostatectomy range from 63–89% and 84–94% respectively [12,14,15]. Whether nerve sparing radical prostatectomy should be attempted in this group of patients is under debate, as the role of nerve sparing radical prostatectomy in high risk prostate cancer patients has only been reported in a few previous studies, with conflicting results [6,12,14,15,16]. Thus, data on the feasibility and efficacy of neurovascular bundle preservation and its effects on survival in high risk prostate cancer patients are limited.

The aim of this study was to assess the role of nerve sparing during open radical prostatectomy in patients with low, intermediate, and high risk prostate cancer and its effects on postoperative functional recovery of urinary continence and erectile function while taking short- to long-term safety outcomes (peri-operative outcomes, disease recurrence, and survival outcomes) into consideration. Comparison of these outcome measures between the two risk groups (low and intermediate vs. high risk) is necessary in order to draw conclusions about the impact of high risk disease on functional and safety outcomes.

## 2. Materials and Methods

### 2.1. Patient Population

In this single-centre cohort study, we reviewed data on 1340 consecutive patients with prostate cancer who underwent open radical prostatectomy with extended pelvic lymph node dissection at our institution from 1996 to 2020. According to the guidelines we used preoperative PSA > 20 ng/mL, pathological biopsy Gleason score ≥ 8, and clinical stage ≥ T3 as the definition of high risk disease. Patients considered inoperable based on staging and digital rectal examination were excluded.

The study was conducted in accordance with the Strengthening the Reporting of Observational Studies in Epidemiology (STROBE) statement and approved by the Ethics Committee of Canton Bern, Switzerland (KEKBE 2016-00156); the need for informed consent was waived.

### 2.2. Selection Criteria

In this study, all patients undergoing open radical prostatectomy with any degree of nerve sparing (no, unilateral, or bilateral nerve sparing) for prostate cancer at our institution from 1996 to 2020 were included. In brief, nerve sparing was performed if there was no ipsilateral palpable induration on digital rectal examination, no ipsilateral capsular involvement in the preoperative MRI, and no contraindication intraoperatively. For inclusion criteria to attempt nerve sparing radical prostatectomy, see Table 1.

### 2.3. Staging and Follow-Up Data Collection

Preoperative staging included physical examination, measurement of PSA, and mpMRI of the pelvis (from 2010 onwards), followed by prostate biopsies. At our department, nearly all patients (excluding those with contraindications such as non-adjustable pacemaker or claustrophobia) underwent mpMRI from late 2010 onwards. Biopsy techniques, including biopsy templates, have changed over the observed study period. Until 2013 we used the ultrasound-guided transrectal approach, followed by the MR/TRUS-fused approach either transrectally (under local anesthetics) or transperineally (under general anesthetics). Given that we are a tertiary referral centre, some patients who had been referred for the radical prostatectomy still underwent the conventional ultrasound-guided transrectal approach in recent years.

Staging also included CT scan of the thorax and abdomen and a whole-body-scintigraphy to exclude bone metastases [17]. All patients were followed prospectively according to the institutional follow-up-protocol following EAU guidelines [6]. Data were prospectively recorded in the institutional database. Histopathological slides were reviewed by an experienced uropathologist. Validated ICIQ short forms and International Index of Erectile Function-15 questionnaires were given to patients preoperatively and at 3, 6, 12, and 24 months. Postoperative PSA measurements and clinical examinations were taken at 3, 6, and 12 months and afterwards yearly. Further diagnostic imaging was obtained with a rising PSA or clinical suspicion.

### 2.4. Functional Outcomes

Urinary continence was defined as complete dryness or occasional loss of no more than a few drops of urine demanding not more than one safety pad per 24 h by self-report or by question 3 and 4 of the ICIQ questionnaire. All patients were continent prior to surgery. Erectile function recovery was defined as the ability to achieve erection sufficient for penetration and maintenance of intercourse by patient self-report or with a score of 3 or more points in question 2 of the IIEF-15-questionnaire with or without erectile aids. Penile rehabilitation with phosphodiesterase type 5 inhibitors three times a week was recommended for all preoperatively potent patients. For inclusion and exclusion criteria for evaluation of urinary continence and erectile function recovery, see Table 2.

### 2.5. Oncological Outcomes

Positive surgical margin was defined as malignant cells at the inked margin of the prostatectomy specimen. The specimens were evaluated for capsular involvement, extra-prostatic spread, and positive surgical margins by senior uropathologists of our center.

Biochemical recurrence was defined as PSA ≥ 0.2 ng/mL in two consecutive measurements [3].

With a rising PSA and/or clinical suspicion, imaging (CT, PET-CT) was obtained to assess the amount and localization of tumor recurrence. We registered and counted separately any metastases, local recurrence, bone metastases, and lymph metastases. Additionally, all adjuvant therapies such as antiandrogenic therapy, chemotherapy, and radiation therapy were recorded.

Local recurrence was defined as tumor recurrence in the prostatic bed, including the areas adjacent to the vesicourethral anastomosis, the rectovesical space, or the seminal vesicle bed.

Overall survival was considered the time from prostatectomy to patient death. Patients who were alive were censored from the last date of consultation.

### 2.6. Surgical Procedure

In essence, the same standardised surgical technique for open nerve sparing radical prostatectomy and pelvic lymph node dissection has been performed for the last 20 years at our centre, as described previously [18,19]. The decision regarding the preoperative administration of anticoagulant agents was made on an individual patient basis [20]. Every radical prostatectomy was performed or supervised by one of three senior surgeons. Nerve sparing was attempted on the non-tumour-bearing side and in patients with nonpalpable tumours on the ipsilateral side if no more than one biopsy core was positive [19]. Furthermore, the final decision to attempt neurovascular bundle preservation was based on preoperative clinical and radiological staging and the intraoperative judgement of tumour localisation and extension. The degree of attempted nerve sparing (no, unilateral, or bilateral) was judged by the surgeon and by inspection of the specimen.

### 2.7. Complications

Complications occurring during follow-up were prospectively entered in our database. For each patient, the presence of any complication, the total number of complications, and the consequential interventions were assessed and graded according to the modified Clavien-Dindo system [21,22].

Grade I to II complications were defined as minor and grade IIIa to V as major. The comprehensive complication index values were then derived from the Clavien-Dindo grades. The Bern comprehensive complication index was calculated using our recently developed modified comprehensive complication index formula [23,24].

### 2.8. Statistical Analysis

To investigate the association of neve sparing and functional outcome in the study cohort, we calculated a propensity score and used inverse probability of treatment weighting (IPTW) to construct balanced treatment groups with respect to risk factors [25,26].

In particular, we conducted two main analyses. First, we included the entire cohort and applied propensity modelling and IPTW to the three treatment groups of patients who had (1) ‘no nerve sparing’ vs. (2) ‘unilateral’ vs. (3) ‘bilateral’ nerve sparing on functional outcomes 24 months after surgery, including the entire study cohort. We estimated the proportion of patients with complete erectile function recovery, erectile function recovery with aid, and continence after no nerve sparing and the average treatment effects of unilateral and bilateral nerve sparing as compared to no nerve sparing after IPTW. We included age, Charlson comorbidity index, diabetes, coronary artery disease, smoking, preoperative pelvic radiation, PSA (log transformed), tumour stage (categorical), positive lymph nodes (binary), pathological lymph node metastases, clinical distant metastases, lymphovascular invasion, clinical T-stage 3 or 4, ISUP 4 or 5, prostate volume (CC), and tumour volume >50% as covariates in the propensity model. An estimated probability ≥ e^−5^ (=0.0067) to receive a treatment was considered sufficient to fulfil the assumption of probability for a treatment. Patients with lower probability for any treatment were excluded from the analysis. To address the question of whether nerve sparing is beneficial in patients with high risk of disease recurrence, we repeated this analysis with risk stratification. To derive the IPT-weighted time-to-event estimates, we carried out pairwise propensity modelling of ’no nerve sparing’ vs. ‘unilateral nerve sparing’ and ‘no nerve sparing’ vs. ‘bilateral nerve sparing’, including the same variables as covariates in the model.

Second, we investigated whether there is a benefit of bilateral as compared to unilateral nerve sparing taking into account the risk of disease recurrence. As such, we dropped patients who did not undergo nerve sparing prostatectomy from this part of the analysis and carried out propensity modelling with IPTW in the remaining sub-cohort, including the same covariates in the analysis as above, first including all these patients, and then stratifying risk category. In patients with low or intermediate risk, some covariates are zero by definition (positive lymph nodes, ISUP 4 or 5, pathological lymph node metastases, clinical distant metastases, PSA, and pathological T-stage < 3). We checked the model fit by plotting kernel density and standardized differences. We analyzed functional outcomes by calculating the IPT-weighted proportions of patients with recovered potency and continence at 3 months, 6 months, 1 year, and 2 years after surgery. To derive treatment effects on binary functional outcome variables within the entire period from 3 months to 2 years of follow-up, we used IPT-weighted population-averaged panel-data models that fit generalized linear estimating equations for calculating binomial regressions, with logit as a link function. All analyses after IPTW applied robust standard errors to account for the weighting. Continuous variables are presented as mean ± standard deviation (SD) with *p* values calculated using linear regression if normally distributed, or as geometric mean with SD and p derived through log-transformation if the distribution was skewed. Categories are shown as numbers and percentages with *p* values from logistic regression or multinomial regression as appropriate. To compare oncological outcomes of unilateral vs. bilateral nerve sparing stratified by high risk, we calculated Kaplan–Meier estimates with corresponding curves using IPT-weighted data.

We used the Kaplan–Meier method and log rank tests to report on oncological outcomes. Statistical analyses were performed using Stata 16 (StataCorp, College Station, TX, USA).

## 3. Results

### 3.1. Baseline Characteristics

A total of 1340 patients were included in this analysis, of which 12% (*n* = 158) underwent non-nerve sparing radical prostatectomy and 39% (*n* = 516) and 50% (*n* = 666) had uni- and bilateral nerve sparing radical prostatectomy, respectively. Demographic, oncological, and surgical characteristics as well as preoperative laboratory values are presented in Table 3 (according to the degree of attempted nerve sparing) and Table 4 (according to the risk profile).

The details of the propensity scores are shown in Figure 1.

### 3.2. Functional Outcomes

#### 3.2.1. Comparison of Patients Having Unilateral vs. Bilateral Nerve Sparing Only

Considering only high risk patients undergoing any nerve sparing, bilateral nerve sparing was associated with a higher probability of erectile function recovery and complete erectile function recovery as compared to unilateral nerve sparing at all time points (see Table 5 and Table 6). SDs were slightly larger before IPTW than afterwards. The same pattern is observed in the requirement of erectile aid for sexual intercourse.

There was no significant benefit of bilateral nerve sparing with respect to urinary continence. As shown in Table 7, patients with low or intermediate risk prostate cancer experienced a similar association at a slightly higher level of recovery, yielding the same SD, keeping in mind that the SD do not depend on sample size whereas the *p* value does.

Table 8 shows ORs with 95% CIs of bilateral vs. unilateral nerve sparing that relate to the entire follow-up period for all functional assessments before and after IPTW. As expected, ORs were slightly higher before IPTW and the largest treatment effect was demonstrated in the group of low or intermediate risk patients. However, in high risk patients, bilateral nerve sparing was associated with increased probability of complete erectile function recovery (OR 1.94, *p* = 0.005) and erectile function recovery (OR 1.53, *p* = 0.010).

Appendix A shows the functional outcomes of uni vs. bilateral nerve sparing 3, 6, 12, and 24 months after prostatectomy in the entire sub-cohort of patients who underwent nerve sparing radical prostatectomy. Bilateral nerve sparing was associated with a higher probability of erectile function recovery and complete erectile function recovery as compared to unilateral nerve sparing at all time points. The SDs were in general slightly larger before IPTW than afterwards. The same pattern was shown in the amount of support needed for sexual intercourse.

#### 3.2.2. Comparison of Patients Having No Nerve Sparing vs. Unilateral vs. Bilateral Nerve Sparing

As shown in Table 9, we found a benefit of unilateral or bilateral nerve sparing over no nerve sparing with respect to functional outcomes at 24 months postoperatively when specifically taking high risk prostate cancer into account. Only *n* = 32 patients had a probability below 0.0067 of receiving one of the three treatments and were consequently excluded from the analysis.

After IPTW, only 3% of patients having non-nerve sparing radical prostatectomy reached complete erectile function recovery (without erectile aid) at 24 months, whereas 22% reached erectile function recovery (with erectile aid) and 87% were continent. Unilateral nerve sparing increased the probability of functional recovery in all outcomes (OR 1.1 or 1.2, respectively), bilateral nerve sparing slightly more so (OR 1.1 to 1.4). ln the sub-cohort of patients with low or intermediate risk, a higher proportion of patients reached erectile function recovery and complete erectile function recovery, corresponding to a lower proportion in the high risk sub-cohort. With respect to treatment effects, we found the same patterns in both strata (low or intermediate risk and high risk) as in the entire cohort. Figure 2a–c shows rates of complete erectile function recovery, erectile function recovery with erectile aid, and continence rates in all groups. Two years after surgery, as depicted in Table 10, functional outcomes showed similar patterns in the low or intermediate risk and high risk patients when stratified by grade of nerve sparing.

### 3.3. Oncological Outcomes

The mean follow-up of the entire cohort (low–intermediate risk and high risk prostate cancer) was 9.7 years (SD 6.2 years), the maximal follow-up was 25 years, and the cumulative follow-up was 13,086 years.

Crude Kaplan–Meier curves showed the worst oncological outcomes in patients who underwent no nerve sparing and best outcomes in patients who underwent bilateral nerve sparing (*p* < 0.001 for all-cause mortality, recurrence, and cancer specific mortality and *p* = 0.034 for mortality from other causes), as expected (see Figure 3). After IPTW, oncological outcomes were similar after no nerve sparing and unilateral nerve sparing (*p* ≥ 0.12) and after no nerve sparing vs. bilateral nerve sparing with respect to recurrence, cancer specific mortality, and mortality from other causes (*p* ≥ 0.059) and were better after bilateral nerve sparing for all-cause mortality (*p* = 0.03). Note that we conducted pairwise propensity modelling (no nerve sparing vs. unilateral nerve sparing and no nerve sparing vs. bilateral nerve sparing) to derive IPT-weighted Kaplan–Meier curves and log rank tests (see Figure 3a–h), and Appendix A.

Survival was favorable in patients undergoing bilateral nerve sparing compared to unilateral nerve sparing (Kaplan–Meier estimate of the entire cohort was 89% and 78%, respectively, after 12 year entire cohort, log rank test: *p* < 0.001). This difference was not observed in patients with low or intermediate risk disease (91% and 88%, *p* = 0.82), only in high risk-patients (86% and 69%, *p* < 0.001). After IPTW, this pattern remained unchanged; see Figure 3d–h. With respect to recurrence, we did not find any difference between unilateral and bilateral nerve sparing after IPTW, as in the entire cohort 48% were event-free after unilateral and 52% after bilateral nerve sparing, *p* = 0.665, low risk 71 and 73%, *p* = 0.693, and high risk 25 and 32%, *p* = 0.26. This could be interpreted in the sense that the propensity model failed to achieve balance with respect to all oncological outcomes. Two years after surgery, oncological outcomes yielded similar patterns in low or intermediate risk and high risk patients when stratified by grade of nerve sparing (see Table 11).

### 3.4. Complications

Bern comprehensive complication index 30-day and 90-day outcomes were similar between the groups. Clavien-Dindo grades were comparable as well (see Appendix A).

## 4. Discussion

Our study shows a statistically significant benefit of unilateral or bilateral nerve sparing over no nerve sparing in both risk groups (low–intermediate risk and high risk prostate cancer) with respect to functional outcomes at 24 months postoperatively, with an ongoing improvement in all three nerve sparing groups during the first two years. Considering only patients undergoing any nerve sparing, bilateral nerve sparing was associated with a higher probability of erectile function recovery as compared to unilateral nerve sparing in both risk groups at all time points. More specifically, in the sub-cohort of patients with low or intermediate risk, a higher proportion of patients reached erectile function recovery, corresponding to a lower proportion in the high risk sub-cohort. However, there was no substantial benefit of bilateral nerve sparing with respect to urinary continence outcomes in either risk group. Importantly, as already demonstrated in our recent study [13], nerve sparing radical prostatectomy can be safely performed without compromising oncological outcomes in patients with high risk disease, who are typically not offered nerve-sparing during surgery because of concerns around inferior cancer prognosis.

Several studies have addressed the impact of nerve sparing on urinary continence and erectile function recovery. A recently published systematic review and meta-analysis [11] reports nerve sparing to be associated to better functional outcomes, whereas others have failed to find such an association [7,8]. However, comparison of functional results in high risk prostate cancer patients is difficult, as data on urinary continence and erectile function recovery after radical prostatectomy in these patients remains scarce. Little is known about outcomes stratified by degree of nerve sparing. Furthermore, comparability is limited by the heterogeneity of definitions of urinary continence, erectile function recovery, and high risk disease. Our functional results (with an overall urinary continence rate of 94% and erectile function recovery rate of 43% at 12 months postoperatively) are consistent with previously reported short- to mid-term urinary continence rates in high risk prostate cancer patients predominantly after robot-assisted radical prostatectomy, ranging from 78% to 100%, and erectile function recovery-rates, ranging from 23% to 70% [12,14,15,16,27,28,29]. Most of these studies mainly focus on the time point 12 months postoperatively. In our longitudinal series, urinary continence and erectile function recovery rates continuously increased after surgery. Hence, patients suffering from urinary incontinence and erectile dysfunction may regain function even after 12 months. This may be due to the ongoing resolution of neurapraxia seen up to 2 years after radical prostatectomy [30]. These encouraging data are important when counseling high risk prostate cancer patients to maintain regular pelvic floor exercises to enhance continence recovery or to continue medical penile rehabilitation after radical prostatectomy.

In line with the current literature, the effect of nerve sparing on erectile function recovery is more pronounced than the effect on urinary continence. This may be explained by the fact that the neurovascular bundle does not contribute to urinary continence as much as it does on erectile function. Other mechanisms may have a larger role to play in continence outcomes, such as preservation of urethral length and the supporting periprostatic structures (e.g., puboprostatic ligaments supporting the external sphincter, etc.). However, we did find that the recovery of urinary continence was significantly higher for nerve sparing compared to bilateral wide resection. This conflicts with the meta-analysis by Reeves et al. [31], which found that nerve sparing did not impact continence rates at 12 and 24 months. On the other hand, other reports have reported that nerve sparing is an important predictor of post-operative continence [5,26,27]. A hypothesis for this observation is that the extent of neurovascular bundle preservation improves the structural or vascular support to the external sphincter and the cavernosal nerve, which seems to be essential in maintaining continence and erectile function recovery. From an anatomical point of view, the fact that some degree of nerve sparing is associated with improved continence is supported by advanced studies by Röthlisberger et al. [32] showing that the continence organ is not only innervated by autonomic nerves providing nervous supply for the rhabdosphincter and lissosphincter, but by somatic nerves which are represented by the pudendal nerve, leave the lesser pelvis, and run in the ischioanal fossa to reach the pelvic floor muscles from the outside of the pelvis. Hence, somatic nerves are not part of the neurovascular bundle and are not affected during radical prostatectomy, partly preserving urinary continence. However, sparing nerve fibres of the exclusively autonomic innervated muscles (Lissosphincter) might keep its function and further improve recovery of urinary continence, indicating that bilateral nerve sparing results in better urinary continence, though without a significant difference from the unilateral nerve sparing group.

Erectile function is dependent on autonomic nerves exclusively delivered by the neurovascular bundles. The small nerve fibers are found right next to the seminal vesicle and lateral from the prostate, running caudal in the direction of the corpora cavernosa and strictly supplying the ipsilateral corpus cavernosum. This may explain the significant difference seen between bilateral and unilateral nerve sparing, which is indicative of a higher impact of neurovascular bundle preservation on erectile function recovery than on urinary continence. Moreover, due to the lateral course of the nerve fibres (within the neurovascular bundle) supplying the corpora cavernosa, some of these fibres may even be spared in patients with partially attempted nerve sparing. Therefore, it is conceivable that sparing at least some of the neurovascular bundle decisively improves erectile function recovery. This is in line with Nguyen et al. [11], who reported that the grade of nerve sparing is associated with higher erectile function recovery. In our study, bilateral nerve sparing was associated with a higher probability of erectile function recovery and complete erectile function recovery as compared to unilateral nerve sparing at all time points. However, a high proportion of men report erectile dysfunction despite nerve sparing radical prostatectomy, which might be explained by decrease of potency over time as well as by diversity in neurovascular bundle anatomy in the posterolateral region of the prostate, leading to unintended damaging of cavernosal nerve fibres during surgery. Furthermore, advanced age and coexisting comorbidities may impede resolution of neurapraxia and vascular recovery [33].

The fact that patients with unilateral nerve sparing may experience the same satisfactory functional outcomes compared to those with attempted bilateral neurovascular bundle preservation implicates other factors critical to functional recovery that have to be taken into account when analyzing differences in urinary continence and erectile function after radical prostatectomy. Surgical techniques have been significantly changed thanks to the advantages of robotics, with many different techniques and approaches (e.g., standard anterior prostatectomy vs. posterior Retzius-sparing radical prostatectomy) developed over the years to improve both functional and oncological outcomes. As such, the Retzius-sparing approach has been evaluated in various studies. The aim of this technique is to spare the anatomical structures surrounding the prostate by approaching the gland posteriorly, thereby keeping the anterior connection between the pubis and the bladder intact [34]. Galfano et al. showed in a large multicentric study that Retzius-sparing radical prostatectomy is feasible and safe in the setting of high-risk prostate cancer. In terms of functional outcomes, two recent systematic reviews of comparative studies reported earlier recovery of urinary continence when performing the Retzius-sparing approach as compared to the standard approach, even in the subpopulation of high-risk patients where early continence is intrinsically harder to achieve due to a more disruptive surgery compared to low-risk disease [35,36]. It is expected that future technological improvements and the creation of a standard training program may further increase the adoption of the Retzius-sparing approach [37].

Furthermore, the reconstruction of supporting structures seems to positively impact functional outcomes as well. In this context, the total anatomical reconstruction technique during anterograde robotic radical prostatectomy described by Manfredi et al. represents a “tension-free” anastomosis technique that aims to restore the anterior and posterior supports to the sphincter. The authors demonstrated excellent results in the early recovery of urinary continence with this technique [38].

With regard to oncological outcomes, several studies have reported conflicting results of nerve sparing radical prostatectomy, as there is an increased risk of a positive surgical margin in high risk prostate cancer, leading to recurrence and progression. Recabal et al. [28] concluded that complete bilateral nerve sparing should not be attempted in high risk prostate cancer patients. Other authors have reported that nerve sparing radical prostatectomy is feasible in selected patients with high risk prostate cancer from an oncological point of view [12,16,39]. Our overall positive surgical margin rates of 29% are consistent with the current literature, ranging from 12–66% in high risk prostate cancer patients. This may be explained by appropriate patient selection and a better understanding of the anatomy of the prostate and its surrounding tissue [32,40]. The 5, 10, and 15 year overall survival and cancer-specific survival of our entire cohort are in agreement with those of previous studies. In our Kaplan–Meier analysis, cancer-specific survival and overall survival were significantly higher according to the grade of nerve sparing, owing to more favorable disease prognosis in those patients having nerve sparing radical prostatectomy.

To decide whether or not a nerve sparing approach should be followed and which grade of nerve sparing to attempt, mpMRI is increasingly being adopted in the prostate cancer clinical pathway. One of the suggested benefits is its excellent ability to identify T3 disease [41]. In this regard, continuous technological progress has been associated with improved oncological and functional outcomes. For instance, based on mpMRI bidimensional images, 3D prostate models have been created for use intraoperatively to allow for modulation of nerve sparing. Checcucci et al. demonstrated limited occurrence of PSM, especially in patients with extracapsular extension at mpMRI or presence of pT3 prostate cancer, on the final histology [42]. On the other hand, Martini et al. described a personalized approach to determine the grade of attempted nerve sparing via automated interaction detection. A machine learning-based partitioning algorithm was applied to identify risk groups by predicting extracapsular extension on final pathology contralaterally to the prostate lobe with clinically high risk disease. They were able to show that wide bilateral excision in men with unilateral high risk disease is not justified, and concluded that full and incremental nerve sparing in cases of contralateral low and intermediate risk for extracapsular extension, respectively, is safe from an oncological stand point [43].

However, cases of discordant clinical (pre- and intraoperative) and imaging findings exist as well. As such, an mpMRI indicating organ-confined disease does not automatically implicate feasibility of nerve sparing. Therefore, intraoperative judgement is necessary as to whether or not the NVBs can be swiped off the prostate with ease (indicating that tumor extension is less likely). If no clear plane can be identified between the prostatic fascia and the NVBs, we advocate for incremental (partial) nerve sparing on the particular side at the maximum. If the NVBs are clearly adherent to the prostate (possibly indicating tumor infiltration into the NVB, capsular infiltration or extraprostatic extension, or significantly fibrosed bundle tissue), then persistent trial to release the bundle off the prostate in a rough manner might cause capsular tear and subsequently positive surgical margins; consequently, nerve sparing should not be attempted.

If the mpMRI is indicative of advanced tumour (cT3-4), we advocate for a non-nerve sparing approach, a careful attempt, or incremental (partial) nerve sparing on the particular tumor-bearing side (in case of capsule bulging only with no signs of extraprostatic extension). Clearly, there should be no full nerve sparing, as the network of nerve fibers and extra-capsular tumor cells are both microscopic and their anatomical structures are hard to recognize visually. Even in robotic cases which magnify the field by 10–12 times, the anatomical structures are difficult to visualize intraoperatively, leading to more possibility of positive surgical margins.

There are several limitations to this study that need to be considered. This was a single-centre retrospective study that is liable to the biases of such analyses and may not be generalizable to wider populations. We employed an IPTW with the aim of balancing the baseline characteristics of patients between the groups; however, this only balances measured variables and cannot account for unmeasured confounders as randomisation is able to. Therefore, there may have been unmeasured confounders that contributed to our results. Furthermore, there was no clearly defined protocol to determine which patients were suitable for nerve sparing or the extent of nerve sparing; thus, these results may not be easily reproducible. The surgeons in this centre were experienced high-volume operators, and it may not be possible for those with less experience to replicate these outcomes. Finally, this was a purely open radical prostatectomy series; hence, comparisons to the conventional laparoscopic and robotic approaches were not possible. However, there are multiple strengths of this cohort, including the long follow-up and large cohort.

## 5. Conclusions

We found that nerve sparing radical prostatectomy in high risk prostate cancer patients is feasible with good functional outcomes without compromising oncological outcomes. Intra-operative preservation of the neurovascular bundle should be considered in carefully selected patients with high risk disease. Future studies should develop risk-stratification tools to identify which high risk prostate cancer cases are suitable for nerve sparing.

## Figures and Tables

**Figure 1 cancers-15-05839-f001:**
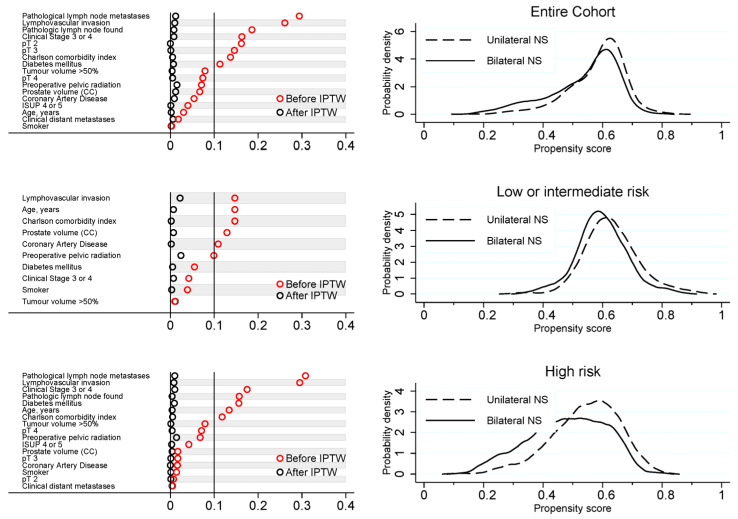
Details of the propensity scores.

**Figure 2 cancers-15-05839-f002:**
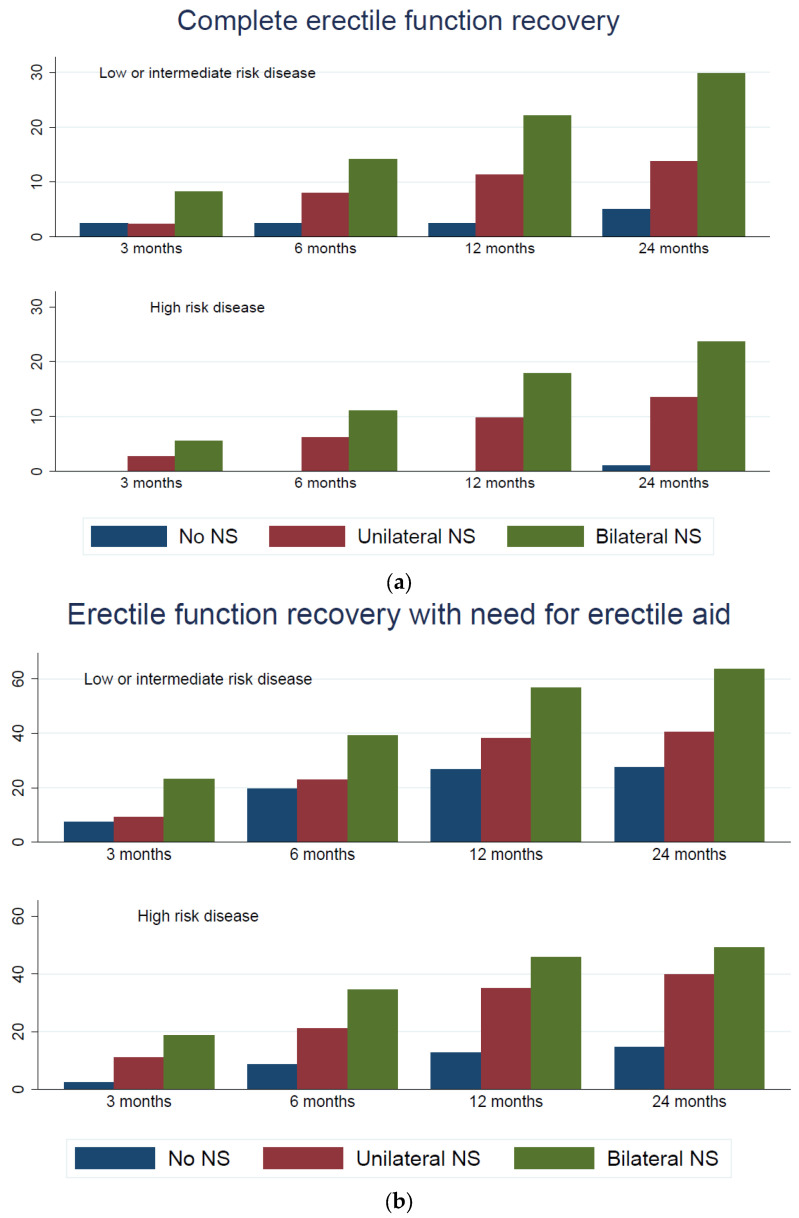
Functional outcomes in low and intermediate vs. high risk prostate cancer according to the risk group and degree of attempted nerve sparing. (**a**) Complete erectile function recovery. (**b**) Erectile function recovery with need for erectile aid. (**c**) Urinary continence.

**Figure 3 cancers-15-05839-f003:**
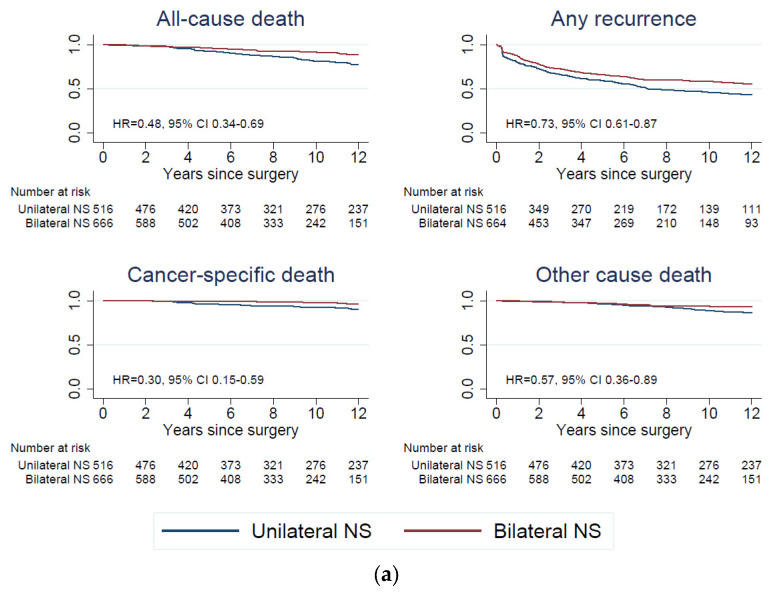
Kaplan-Meier curves of oncological endpoints before and after IPTW: cumulative incidence of oncological. Outcomes (all-cause death, cancer-specific death, other cause death, and any recurrence) and treatment. (**a**) Oncological outcomes of the entire cohort (low-, intermediate- and high risk patients) before IPTW according to the nerve sparing status. (**b**) Oncological outcomes of low-risk patients before IPTW according to the nerve sparing status. (**c**) Oncological outcomes of high-risk patients before IPTW according to the nerve sparing status. (**d**) Oncological outcomes of the entire cohort after IPTW according to the nerve sparing status. (**e**) Oncological outcomes of low-risk patients after IPTW according to the nerve sparing status. (**f**) Oncological outcomes of high-risk patients after IPTW according to the nerve sparing status. (**g**) Oncological outcomes of the entire cohort after IPTW according to the nerve sparing status (no vs. unilateral). (**h**) Oncological outcomes of the entire cohort after IPTW according to the nerve sparing status (no vs. bilateral).

**Table 1 cancers-15-05839-t001:** Inclusion criteria to attempt nerve sparing radical prostatectomy.

Exclusion Criteria for NS	Inclusion Criteria for Unilateral NS	Inclusion Criteria for Bilateral NS
Bilateral palpable induration side (on digital rectal examination)	palpable induration only on non-NS side (on digital rectal examination)	no palpable induration on either side (on digital rectal examination)
bilateral capsular involvement (preoperative MRI)	ipsilateral capsular involvement (preoperative MRI) only on the non-NS side	no ipsilateral capsular involvement in the preoperative MRI
Contraindication intraoperatively for any NS adherence of both NVBs to the prostate (indicating tumor infiltration) bilaterally fibrosed NVB	Contraindication intraoperatively for bilateral NS adherence of the NVB to the prostate (indicating tumor infiltration into the NVB) Fibrosed NVB (on the non-NS side)	no contraindication intraoperatively unresistant bilateral NVB release no visual clues for non-organ confined disease on either side
Bulky disease (i.e., cT4)		

NS, nerve sparing; NVB, neurovascular bundle.

**Table 2 cancers-15-05839-t002:** Inclusion and exclusion criteria used in the analysis of urinary continence and erectile function recovery.

Continence:
*Inclusion criteria:*
Follow-up of at least 3 months post RP Fully continent preoperatively
*Exclusion criteria:*
Radiotherapy to the pelvis prior to RP, and within 24 months Death within 24 months
Erectile function recovery:
*Inclusion criteria:*
Follow-up of at least 3 months
*Exclusion criteria:*
Absence of erection sufficient for penetration prior to RP Radiotherapy to the pelvis with or without androgen-deprivation therapy prior to RP, and within 24 months Androgen-deprivation therapy within 24 months Death within 24 months

**Table 3 cancers-15-05839-t003:** Baseline characteristics of 1340 patients undergoing RP with and without attempted NS.

	No NS	Unilateral NS	Bilateral NS	*p*-Value
Number of patients (%)	158 (12)	516 (39)	666 (50)	
** *Preoperative* **				
Age [years], mean (SD)	65 ± 6.4	64 ± 6.4	64 ± 6.7	0.18
BMI [kg/m^2^], mean (SD)	27 ± 4.2	27 ± 3.9	27 ± 3.8	0.50
CACI, mean (SD)	2.94 ± 3.2	2.78 ± 3.1	2.64 ± 3.0	0.002
Diabetes mellitus, *n* (%)	18 (11)	50 (10)	44 (6.6)	0.048
Coronary artery disease, *n* (%)	17 (11)	51 (10)	77 (12)	0.66
Nicotine, *n* (%)	60 (38)	128 (25)	166 (25)	0.003
Erectile dysfunction, *n* (%)	30 (19)	74 (14)	109 (16)	0.38
Radiation to the pelvis, *n* (%)	2 (1.3)	5 (0.97)	12 (1.8)	0.54
PSA, µg/L	13 [7.7–27]	10 [6.3–16]	7.9 [5.1–13]	<0.001
Prostate volume [cc], mean (SD)	53 ± 20	51 ± 25	52 ± 30	0.45
Digital Rectal Examination, *n* (%)				<0.001
cT1	31 (20)	172 (33)	324 (49)	
cT2	94 (59)	298 (58)	310 (47)	
cT3	30 (19)	46 (8.9)	32 (4.8)	
cT4	3 (1.9)	0 (0.00)	0 (0.00)	
Biopsy ISUP, *n* (%)				<0.001
1	37 (23)	182 (35)	232 (35)	
2	32 (20)	136 (26)	232 (35)	
3	22 (14)	56 (11)	100 (15)	
4	15 (9.5)	51 (10)	60 (9.0)	
5	21 (13)	36 (7.0)	22 (3.3)	
** *Intraoperative* **				
Duration of surgery [min], mean (SD)	241 ± 52	245 ± 49	254 ± 57	0.003
Blood loss [l], median (IQR)	1 [0.7–1.5]	0.8 [0.6–1.2]	0.6 [0.4–0.9]	<0.001
Number of lymph nodes removed, median (SD)	27 ± 12	25 ± 11	28 ± 14	<0.001
** *Postoperative* **				
Prostate volume [cc]	49 [40–60]	46 [37–57]	46 [36–60]	0.030
Tumor volume (%)	23 [23–30]	13 [10–20]	10 [5.0–20]	<0.001
Tumor volume > 50%, *n* (%)	21 (13)	27 (5.2)	24 (3.6)	<0.001
Tumor pathology, *n* (%)				<0.001
pT2	64 (41)	298 (58)	437 (66)	
pT3a	29 (18)	105 (20)	141 (21)	
pT3b	57 (36)	104 (20)	82 (12)	
pT4	8 (5)	9 (2)	6 (1)	
Pathological ISUP, *n* (%)				<0.001
1	49 (31)	183 (35)	167 (25)	
2	29 (18)	153 (30)	280 (42)	
3	26 (16)	72 (14)	106 (16)	
4	16 (10)	55 (11)	63 (9.5)	
5	38 (24)	53 (10)	50 (7.5)	
Lymph node metastases, *n* (%)	50 (32)	128 (25)	89 (13)	<0.001
ENE+, *n* (%)	10 (6.3)	28 (5.4)	42 (6.3)	0.80
Lymphovascular invasion, *n* (%)	78 (49)	173 (34)	146 (22)	<0.001
PSM, *n* (%)	68 (43)	199 (39)	221 (33)	0.030
Prostatitis, *n* (%)	38 (24)	177 (34)	194 (29)	0.028

NS, nerve sparing; IPTW, BMI, body mass index; CACI, Charlson Comorbidity Index; PSA, Prostate-Specific Antigen; ISUP, International Society of Urological Pathology; ENE, extranodal extension; PSM, Positive Surgical Margins.

**Table 4 cancers-15-05839-t004:** Baseline characteristics of 1340 patients undergoing RP with low–intermediate and high risk prostate cancer.

	Total	Low-Intermediate Risk	High Risk	*p*-Value
Number of patients (%)	1340 (100)	614 (46)	726 (54)	
** *Preoperative* **				
Age [years], mean (SD)	64 ± 6.5	63 ± 6.5	65 ± 6.4	<0.001
BMI [kg/m^2^], mean (SD)	27 ± 3.9	27 ± 4.0	27 ± 3.8	0.14
CACI, mean (SD)	0.73 ± 1.1	0.67 ± 1.0	0.77 ± 1.1	0.08
Diabetes mellitus, *n* (%)	112 (8.4)	46 (7.5)	66 (9.1)	0.32
Coronary artery disease, *n* (%)	145 (11)	71 (12)	74 (10)	0.43
Nicotine, *n* (%)	354 (26)	151 (25)	203 (28)	0.17
Erectile dysfunction, *n* (%)	213 (16)	93 15)	120 (17)	0.50
Radiation to the pelvis, *n* (%)	19 (1.4)	7 (1.1)	12 (1.7)	0.49
PSA, µg/L	9.0 [5.7–15]	7.0 [4.7–10]	13 [7.6–23]	<0.001
Prostate volume [cc], mean (SD)	52 ± 27	51 ± 28	52 ± 26	0.37
Digital Rectal Examination, *n* (%)				<0.001
cT1	527 (39)	300 (49)	227 (31)	
cT2	702 (52)	302 (49)	400 (55)	
cT3	108 (8.1)	12 (2.0)	96 (13)	
cT4	3 (0.22)	0 (0.00)	3 (0.41)	
Biopsy ISUP, *n* (%)				<0.001
1	451 (34)	312 (51)	139 (19)	
2	400 (30)	210 (34)	190 (26)	
3	178 (13)	56 (9.1)	122 (17)	
4	126 (9.4)	5 (0.81)	121 (17)	
5	79 (5.9)	1 (0.16)	78 (11)	
** *Intraoperative* **				
Duration of surgery [min], mean (SD)	249 ± 54	249 ± 56	249 ± 52	0.94
Blood loss [dl], median (IQR)	7 [5–11]	7 [5–10]	8 [5–12]	<0.001
Number of lymph nodes removed, median (SD)	27 ± 13	25 ± 13	28 ± 13	<0.001
** *Postoperative* **				
Prostate volume [cc]	46 [37–60]	46 [35–59]	46 [38–60]	0.033
Tumor volume (%)	13 [8–22]	10 [5–15]	15 [10–25]	<0.001
Tumor volume > 50, *n* (%)	72 (5)	10 (2)	62 (9)	<0.001
Tumor pathology, *n* (%)				<0.001
pT2	799 (60)	613 (100)	186 (26)	
pT3a	275 (21)	0 (0)	275 (38)	
pT3b	243 (18)	0 (0)	243 (33)	
pT4	23 (2)	0 (0)	23 (3)	
Pathological ISUP, *n* (%)				<0.001
1	399 (30)	280 (46)	119 (16)	
2	462 (34)	268 (44)	194 (27)	
3	204 (15)	66 (11)	138 (19)	
4	134 (10)	0 (0.00)	134 (18)	
5	141 (11)	0 (0.00)	141 (19)	
Lymph node metastases, *n* (%)	267 (20)	0 (0)	267 (37)	<0.001
ENE+, *n* (%)	80 (6)	0 (0)	80 (11)	<0.001
Lymphovascular invasion, *n* (%)	397 (30)	18 (3)	379 (52)	<0.001
PSM, *n* (%)	488 (36)	160 (26)	328 (45)	<0.001
Prostatitis, *n* (%)	409 (31)	194 (32)	215 (30)	0.44

NS, nerve sparing; IPTW, BMI, body mass index; CACI, Charlson Comorbidity Index; PSA, Prostate-Specific Antigen; ISUP, International Society of Urological Pathology; ENE, extranodal extension; PSM, Positive Surgical Margins.

**Table 5 cancers-15-05839-t005:** Functional outcomes in patients with uni- vs. bilateral NS-RP for high risk prostate cancer before and after IPTW.

	*Before IPTW*				*After IPTW*			
	Unilateral NS	Bilateral NS	Diff.	*p*-Value	Unilateral NS	Bilateral NS	Diff.	*p*-Value
Number of patients	297	312			297	312		
** *Complete EFR, n (%)* **								
3 months	8 (3)	17 (6)	−0.141	0.090	10 (3)	18 (6)	−0.109	0.242
6 months	18 (6)	35 (11)	−0.178	0.033	20 (7)	36 (11)	−0.158	0.079
12 months	29 (10)	56 (18)	−0.236	0.005	30 (10)	55 (18)	−0.218	0.014
24 months	40 (14)	74 (24)	−0.261	0.003	39 (13)	73 (23)	−0.268	0.003
** *EFR, n (%)* **								
3 months	33 (11)	59 (19)	−0.215	0.009	33 (11)	58 (18)	−0.210	0.015
6 months	63 (21)	108 (35)	−0.304	<0.001	66 (22)	109 (35)	−0.283	0.001
12 months	104 (35)	143 (46)	−0.222	0.008	110 (37)	141 (45)	−0.166	0.056
24 months	118 (40)	153 (49)	−0.187	0.032	121 (41)	150 (48)	−0.150	0.095
** *Continence, n (%)* **								
3 months	246 (83)	272 (87)	−0.119	0.147	246 (83)	269 (86)	−0.096	0.264
6 months	267 (90)	288 (92)	−0.081	0.329	268 (90)	288 (92)	−0.068	0.426
12 months	283 (95)	292 (94)	0.070	0.414	283 (95)	292 (94)	0.083	0.339
24 months	290 (98)	299 (96)	0.087	0.342	289 (97)	300 (96)	0.070	0.466
** *Aid, n (%)* **								
3 months				0.001				0.001
*no aid*	13 (4)	21 (7)	0.105		15 (5)	22 (7)	0.093	
*oral PDE-5 inhibitors*	15 (5)	39 (13)	0.268		14 (5)	39 (12)	0.273	
*non-oral (MUSE + ISI)*	11 (4)	3 (1)	−0.182		10 (3)	3 (1)	−0.175	
*erectile dysfunction*	258 (87)	248 (80)	−0.194		258 (87)	248 (80)	−0.198	
6 months				<0.001				0.001
*no aid*	21 (7)	37 (12)	0.158		23 (8)	39 (12)	0.156	
*oral PDE-5 inhibitors*	23 (8)	58 (19)	0.322		24 (8)	56 (18)	0.295	
*non-oral (MUSE + ISI)*	22 (8)	18 (6)	−0.065		23 (8)	21 (7)	−0.045	
*erectile dysfunction*	230 (77)	199 (64)	−0.305		227 (76)	197 (63)	−0.293	
12 months				<0.001				0.005
*no aid*	31 (10)	57 (18)	0.222		32 (11)	57 (18)	0.216	
*oral PDE-5 inhibitors*	38 (13)	66 (21)	0.224		42 (14)	63 (20)	0.163	
*non-oral (MUSE + ISI)*	38 (13)	24 (8)	−0.176		40 (13)	25 (8)	−0.168	
*erectile dysfunction*	190 (64)	166 (53)	−0.221		184 (62)	167 (53)	−0.172	
24 months				0.001				0.003
*no aid*	40 (14)	76 (24)	0.279		39 (13)	76 (24)	0.291	
*oral PDE-5 inhibitors*	35 (12)	51 (16)	0.133		37 (13)	47 (15)	0.071	
*non-oral (MUSE + ISI)*	45 (15)	29 (9)	−0.179		47 (16)	30 (10)	−0.182	
*erectile dysfunction*	178 (60)	157 (50)	−0.194		174 (59)	159 (51)	−0.155	

NS-RP, nerve sparing radical prostatectomy; IPTW, EFR, erectile function recovery; PDE-5, Phosphodiesterase 5: MUSE, medicated urethral system for erection: ISI, intracavernous self-injection with prostaglandin.

**Table 6 cancers-15-05839-t006:** Average treatment effect for bilateral nerve sparing at 24 months postoperatively after IPTW.

	Uni NS	Bi NS	*Excluded **		
	Proportion after IPTW	Average Treatment Effect	*p*-Value	No NS	Uni NS
** *Entire cohort* **				*n* = 516	*n* = 666
Complete EFR	13 (10–16)	1.1 (1.1–1.2)	<0.001	0	0
EFR	40 (36–45)	1.2 (1.1–1.2)	<0.001	0	0
Continence	97 (96–99)	1.0 (1.0–1.0)	0.803	2	0
** *Low or intermediate risk* **				*n* = 219	*n* = 354
Complete EFR	15 (10–19)	1.1 (1.1–1.2)	<0.001	0	0
EFR	42 (35–48)	1.2 (1.1–1.3)	<0.001	0	0
Continence	97 (95–99)	1.0 (1.0–1.0)	0.613	0	0
** *High risk* **				*n* = 297	*n* = 312
Complete EFR	13 (9–17)	1.1 (1.0–1.2)	0.002	0	0
EFR	41 (35–47)	1.1 (1.0–1.2)	0.110	0	0
Continence	97 (95–99)	1.0 (1.0–1.0)	0.544	2	0

* Some patients had to be excluded from the analysis because the estimated probability to receive one of the treatments was below the cutoff (0.0067).

**Table 7 cancers-15-05839-t007:** Functional outcome in patients with uni- vs. bilateral NS-RP for low and intermediate risk prostate cancer before and after IPTW.

	*Before IPTW*				*After IPTW*			
	Unilateral NS	Bilateral NS	Diff.	*p*-Value	Unilateral NS	Bilateral NS	Diff.	*p*-Value
Number of patients	219	354			219	354		
** *Complete EFR, n (%)* **								
3 months	5 (2)	29 (8)	−0.269	0.006	6 (3)	28 (8)	−0.239	0.023
6 months	17 (8)	50 (14)	−0.200	0.028	17 (8)	49 (14)	−0.197	0.034
12 months	25 (11)	78 (22)	−0.292	0.002	24 (11)	76 (22)	−0.292	0.002
24 months	30 (14)	106 (30)	−0.396	<0.001	30 (14)	102 (29)	−0.379	<0.001
** *EFR, n (%)* **								
3 months	20 (9)	82 (23)	−0.390	<0.001	21 (9)	80 (23)	−0.362	<0.001
6 months	50 (23)	139 (39)	−0.356	<0.001	51 (23)	135 (38)	−0.323	<0.001
12 months	84 (38)	201 (57)	−0.378	<0.001	85 (39)	197 (56)	−0.342	<0.001
24 months	88 (40)	225 (64)	−0.478	<0.001	90 (41)	221 (62)	−0.439	<0.001
** *Continence, n (%)* **								
3 months	184 (84)	301 (85)	−0.030	0.731	185 (85)	302 (85)	−0.018	0.835
6 months	194 (89)	330 (93)	−0.161	0.061	196 (90)	331 (94)	−0.143	0.091
12 months	207 (94)	345 (98)	−0.167	0.055	208 (95)	346 (98)	−0.150	0.076
24 months	211 (97)	346 (98)	−0.070	0.436	213 (97)	346 (98)	−0.043	0.614
** *Aid, n (%)* **								
3 months				0.001				0.003
*no aid*	6 (3)	32 (9)	0.271		7 (3)	30 (9)	0.240	
*oral PDE-5 inhibitors*	9 (4)	36 (10)	0.238		9 (4)	35 (10)	0.230	
*non-oral (MUSE + ISI)*	6 (3)	15 (4)	0.082		6 (3)	15 (4)	0.076	
*erectile dysfunction*	198 (90)	271 (76)	−0.382		197 (90)	273 (77)	−0.353	
6 months				<0.001				0.002
*no aid*	18 (8)	53 (15)	0.208		18 (8)	52 (15)	0.203	
*oral PDE-5 inhibitors*	16 (7)	60 (17)	0.289		17 (8)	58 (16)	0.269	
*non-oral (MUSE + ISI)*	16 (7)	27 (8)	0.002		17 (8)	26 (7)	−0.024	
*erectile dysfunction*	168 (77)	214 (61)	−0.351		167 (76)	218 (62)	−0.317	
12 months				<0.001				<0.001
*no aid*	25 (11)	78 (22)	0.292		24 (11)	76 (22)	0.292	
*oral PDE-5 inhibitors*	25 (11)	84 (24)	0.328		25 (12)	82 (23)	0.311	
*non-oral (MUSE + ISI)*	34 (16)	39 (11)	−0.133		36 (16)	38 (11)	−0.161	
*erectile dysfunction*	135 (62)	153 (43)	−0.378		134 (61)	157 (44)	−0.342	
24 months				<0.001				<0.001
*no aid*	30 (14)	106 (30)	0.396		30 (14)	102 (29)	0.379	
*oral PDE-5 inhibitors*	25 (11)	84 (24)	0.328		24 (11)	83 (23)	0.333	
*non-oral (MUSE + ISI)*	33 (15)	36 (10)	−0.154		36 (16)	36 (10)	−0.185	
*erectile dysfunction*	131 (60)	129 (36)	−0.478		129 (59)	133 (38)	−0.439	

NS-RP, nerve sparing radical prostatectomy; IPTW, EFR, erectile function recovery; PDE-5, Phosphodiesterase 5: MUSE, medicated urethral system for erection: ISI, intracavernous self-injection with prostaglandin.

**Table 8 cancers-15-05839-t008:** Average treatment effect for bilateral nerve sparing within follow-up time (3–24 months postoperatively) before and after IPTW.

	Before IPTW		After IPTW	
	OR (95% CI)	*p*-Value	OR (95% CI)	*p*-Value
** *Entire cohort* **				
Complete EFR	2.37 (1.77–3.17)	<0.001	2.32 (1.68–3.21)	<0.001
EFR	2.03 (1.64–2.52)	<0.001	1.88 (1.49–2.36)	<0.001
Continence	1.40 (1.03–1.91)	0.030	1.38 (0.98–1.95)	0.067
** *Low or intermediate risk* **				
Complete EFR	2.74 (1.78–4.22)	<0.001	2.72 (1.68–4.40)	<0.001
EFR	2.41 (1.76–3.30)	<0.001	2.20 (1.57–3.08)	<0.001
Continence	1.19 (0.77–1.85)	0.437	1.18 (0.71–1.95)	0.519
** *High risk* **				
Complete EFR	2.00 (1.33–2.99)	0.001	1.94 (1.23–3.07)	0.005
EFR	1.66 (1.24–2.24)	0.001	1.53 (1.11–2.12)	0.010
Continence	1.70 (1.10–2.64)	0.017	1.53 (0.94–2.50)	0.087

**Table 9 cancers-15-05839-t009:** Functional outcome 24 months postoperatively of all three treatment groups after IPTW.

	No NS	Uni vs. No NS		Bi vs. No NS		Excluded ***		
	Proportion After IPTW	Average Treatment Effect	*p*-Value	Average Treatment Effect	*p*-Value	No NS	Uni NS	Bi NS
** *Entire cohort* **						*n* = 158	*n* = 516	*n* = 666
Complete EFR	3 (0–7)	1.1 (1.1–1.2)	<0.001	1.2 (1.2–1.3)	<0.001	0	0	0
EFR	22 (14–31)	1.2 (1.1–1.3)	<0.001	1.4 (1.2–1.5)	<0.001	0	0	0
Continence	87 (80–93)	1.1 (1.0–1.2)	0.003	1.1 (1.0–1.2)	0.003	2	2	0
** *Low or intermediate risk* **						*n* = 41	*n* = 219	*n* = 354
Complete EFR	5 (0–11)	1.1 (1.0–1.2)	0.009	1.3 (1.2–1.4)	<0.001	0	4	5
EFR	26 (12–40)	1.2 (1.0–1.4)	0.040	1.4 (1.2–1.7)	<0.001	0	4	5
Continence	84 (74–95)	1.1 (1.0–1.3)	0.016	1.1 (1.0–1.3)	0.011	0	4	6
** *High risk* **						*n* = 117	*n* = 297	*n* = 312
Complete EFR	1 (0–3)	1.1 (1.1–1.2)	<0.001	1.2 (1.2–1.3)	<0.001	0	0	0
EFR	21 (10–31)	1.2 (1.1–1.4)	0.002	1.3 (1.1–1.5)	<0.001	0	0	0
Continence	88 (80–96)	1.1 (1.0–1.2)	0.045	1.1 (1.0–1.2)	0.037	2	4	7

* Some patients had to be excluded from the analysis because the estimated probability to receive one of the treatments was below the cutoff (0.0067).

**Table 10 cancers-15-05839-t010:** Functional outcomes as observed (no adjustment) after inverse probability of treatment weighing stratified by grade of nerve sparing and risk group.

Urinary Continence 24 Months Postoperatively				
	No NS	Unilateral NS	Bilateral NS	*p* value
Low and intermediate risk, *n* (%)	31 (82%)	196 (97%)	299 (98%)	<0.001
High risk, *n* (%)	73 (85%)	236 (98%)	241 (96%)	<0.001
	**No NS**	**Any NS**		***p* value**
Low and intermediate risk, *n* (%)	31 (82%)	495 (97%)		<0.001
High risk, *n* (%)	73 (85%)	477 (97%)		<0.001
**Complete erectile function recovery 24 months postoperatively**				
	**No NS**	**Unilateral NS**	**Bilateral NS**	***p* value**
Low and intermediate risk, *n* (%)	2 (5.0%)	28 (14%)	91 (30%)	<0.001
High risk, *n* (%)	1 (1.1%)	36 (14%)	64 (24%)	<0.001
	**No NS**	**Any NS**		***p* value**
Low and intermediate risk, *n* (%)	2 (5.0%)	119 (23%)		0.005
High risk, *n* (%)	1 (1.1%)	100 (19%)		<0.001
**Erectile function recovery 24 months postoperatively**				
	**No NS**	**Unilateral NS**	**Bilateral NS**	***p* value**
Low and intermediate risk, *n* (%)	11 (28%)	82 (40%)	194 (64%)	<0.001
High risk, *n* (%)	14 (15%)	106 (40%)	133 (49%)	<0.001
	**No NS**	**Any NS**		***p* value**
Low and intermediate risk, *n* (%)	11 (28%)	276 (54%)		0.001
High risk, *n* (%)	14 (15%)	239 (45%)		<0.001

**Table 11 cancers-15-05839-t011:** Survival data as observed (no adjustment) stratified by grade of nerve sparing and risk group.

Recurrence-Free Survival 10 Years Postoperatively			
	No NS	Unilateral NS	Bilateral NS
Low and intermediate risk, % (CI)	68 (50 to 81)	74 (67 to 80)	75 (70 to 80)
High risk, % (CI)	25 (17 to 35)	24 (19 to 30)	40 (34 to 46)
	**No NS**	**Any NS**	
Low and intermediate risk, *n* (%)	68 (50 to 81)	74 (70 to 78)	
High risk, *n* (%)	25 (17 to 35)	32 (27 to 36)	
**Cancer-specific survival 10 years postoperatively**			
	**No NS**	**Unilateral NS**	**Bilateral NS**
Low and intermediate risk, *n* (%)	97 (81 to 100)	99 (96 to 100)	100 (100 to 100)
High risk, *n* (%)	80 (69 to 87)	87 (81 to 91)	96 (92 to 98)
	**No NS**	**Any NS**	
Low and intermediate risk, *n* (%)	97 (81 to 100)	100 (98 to 100)	
High risk, *n* (%)	80 (69 to 87)	91 (88 to 93)	
**Overall survival 10 years postoperatively**			
	**No NS**	**Unilateral NS**	**Bilateral NS**
Low and intermediate risk, *n* (%)	91 (75 to 97)	92 (87 to 95)	93 (89 to 95)
High risk, *n* (%)	69 (58 to 78)	73 (67 to 79)	90 (85 to 94)
	**No NS**	**Any NS**	
Low and intermediate risk, *n* (%)	91 (75 to 97)	92 (89 to 94)	
High risk, *n* (%)	69 (58 to 78)	81 (77 to 85)	

## Data Availability

Data can be requested from the corresponding author with corresponding approval by the local ethics committee of the requesting institution.

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
