# Peer review of "Functional Impact of Neuro-Vascular Bundle Preservation in High Risk Prostate Cancer without Compromising Oncological Outcomes: A Propensity-Modelled Analysis"

_cancers, 2023, doi:10.3390/cancers15245839_

Round 1

Reviewer 1 Report

Comments and Suggestions for Authors

Congratulate the researcher team for the initiative in this area. This topic is very interesting and work like this is needed to find solid results.

I think it is an important first step, furthermore, I consider that the design is adequate and reflects what the researchers are trying to find,

It has weaknesses like many studies, being truly retrospective, although the cohort is prospective and consecutive; Also the fact of being single-center and not being compared with new techniques such as laparoscopy and robotics, something that is essential today.

It would be interesting to reflect in a summary table which patients you consider benefit from preservation, on the one hand unilateral and on the other bilateral, to take into account when applying your findings.

I have two issues that are not clear to me. As a high-risk patient, did everyone have an MRI prior to surgery? When was it done? According to what they always refer to pre-biopsy, but at that time this was not indicated? how this was interpreted when deciding preservation.

You take into account the PSA nadir, I have not seen it reflected. These are very important when evaluating the overall disease and prognosis and more of these pa

Author Response

1. I congratulate the researcher team for the initiative in this area. This topic is very interesting and work like this is needed to find solid results. I think it is an important first step, furthermore, I consider that the design is adequate and reflects what the researchers are trying to find. It has weaknesses like many studies, being truly retrospective, although the cohort is prospective and consecutive.
Reply: We thank the reviewer and greatly appreciate this kind comment.

2. Also the fact of being single-center and not being compared with new techniques such as laparoscopy and robotics, something that is essential today.
Reply: This is an important suggestion. We have adapted the limitations in the discussion part accordingly and added that our cohort is a purely open RP series and hence comparison to the conventional laparoscopic and robotic approach is not possible.
3. It would be interesting to reflect in a summary table which patients you consider benefit from preservation, on the one hand unilateral and on the other bilateral, to take into account when applying your findings.
Reply: We are grateful for this recommendation. We have created a new Table 1a ‘Inclusion criteria to attempt nerve sparing radical prostatectomy’. Table 1 has been renamed to Table 1b and 1c.

4. I have two issues that are not clear to mäe. As a high-risk patient, did everyone have an MRI prior to surgery? When was it done? According to what they always refer to pre-biopsy, but at that time this was not indicated? how this was interpreted when deciding preservation.
Reply: We appreciate these questions and we aim at replying to them seperately.
Given multiparametric MRI (mpMRI) became only recognised in 2010, patients until then did only undergo digito-rectal exam and prostate biopsy for local staging. At our department nearly all patients (excluding those with contraindications such as non-adjustable pacemaker or claustrophobia) underwent mpMRI from late 2010 onwards. Up until 2013 we have performed conventional TRUS-biopsies, afterwards the diagnostic pathway changed to the MR/TRUS-fused approach, either transrectally (under local anesthetics) or transperineally (under general anesthetics) with some surgeons performing combined template and targeted biopsies, or template or targeted biopsies only.
This has been specified in the section ‘2.3. Staging, follow-up data collection’ of the manuscript.
To answer to the last part of this question comprehensively, the differentiation needs to be made whether the imaging data are more suspicious for more advanced disease (i.e. capsule bulging indicating infiltration, or extraprostatic growth) than clinical pre- and intraoperative findings, or vice versa (i.e. digito-rectal exam or intraoperative findings suspicious for more advanced cancer but organ-confined tumor without evidence for extraprostatic extension on MRI).
In case of image morphology indicating advanced tumor we advocate for a wide excision (no nerve sparing) or incremental (partial) nerve sparing on the particular tumor bearing side. We do not perform full nerve sparing in this situation as the network of nerve fibers and extra-capsular tumor cells are both microscopic, thus their anatomical structures are hard to be recognized visually. Even in robotic cases which magnifies the field by 10–12 times, their anatomical structures are still difficult to visualize intraoperatively, leading to more possibility of positive surgical margins.
On the other hand, clinically negative MRI (i.e. cleary indicating organ-confined disease) does not automatically implicate feasibility of nerve sparing. As such, in case of digito-rectal exam suspicious for more advanced disease despite less concerning MR-findings cancer control and functional outcomes should be balanced based on the intraoperative judgement if the neurovascular bundles can be swiped off the prostate with ease (indicating that tumor extension is less likely) or not (as it is clearly adherent to the prostate which may indicate tumor infiltration into the neurovascular bundle, and hence being indicative for capsular infiltration or extraprostatic extension, or significantly fibrosed bundle tissue in which case persistent trial to release the bundle off the prostate in a rough manner might cause capsular tear and subsequently positive surgical margins). Conclusively, if no clear plane can be identified between the prostatic fascia and the neurovascular bundles we advocate for an incremental (partial) nerve sparing on the particular side.
This has been added to the discussion of the manuscript.
5. You take into account the PSA nadir, I have not seen it reflected. These are very important when evaluating the overall disease and prognosis and more of these pa
Reply: Initially, PSA measurement as part of the follow-up after radical prostatectomy is performed at 3, 6 and 12 months and afterwards yearly. As such, PSA nadir should be in the non-detectable range after 3 months. Further diagnostic imaging was obtained with a rising PSA or clinical suspicion. Given we specifically focus on the functional outcomes we did not evaluate PSA-persistence after radical prostatectomy but we focused extensively on oncological outcomes in our recent publication, please see also Furrer MA, Sathianathen N, Gahl B, Corcoran NM, Soliman C, Rodriguez Calero JA, Ineichen GB, Gahl M, Kiss B, Thalmann GN. Oncological outcomes after attempted nerve-sparing radical prostatectomy (NSRP) in patients with high-risk prostate cancer are comparable to standard non-NSRP: a longitudinal long-term propensity-matched single-centre study. BJU Int. 2023 Aug 7. doi: 10.1111/bju.16126. Epub ahead of print. PMID: 37548822.

Reviewer 2 Report

Comments and Suggestions for Authors

Dear Authors,

I read with interest your manuscript entitles: Functional impact of neuro-vascular bundle preservation in high risk prostate cancer without compromising oncological outcomes: a propensity-matched analysis”. 

Overall, I think this article is very interesting. 

I think the document has some minor limitations.

Technological advancement has allowed for better and better surgical (oncologic and functional) outcomes. For this reason, the following papers that I recommend you read and cite in your work are important discussion points

-       doi: 10.1007/s00345-022-04038-8. Epub 2022 Jul 5. PMID: 35790535.

-       doi: 10.1097/JU.0000000000002205. Epub 2021 Sep 22. PMID: 34547922.

In addition, beyond nerve sparing other factors are critical to functional recovery such as the type of approach (for example retzius-sparing approach) and the reconstruction of supporting structures. In this regard I advise you to add in your article:

-       doi: 10.23736/S2724-6051.22.05179-5. PMID: 36629812.

doi: 10.1111/bju.14716. Epub 2019 Mar 15. PMID: 30801887

Author Response

1. Dear Authors, I read with interest your manuscript entitles: “Functional impact of neuro-vascular bundle preservation in high risk prostate cancer without compromising oncological outcomes: a propensity-matched analysis”. Overall, I think this article is very interesting.
Reply: We greatly appreciate the reviewer’s kindness and positive feedback.
2. Technological advancement has allowed for better and better surgical (oncologic and functional) outcomes. For this reason, the following papers that I recommend you read and cite in your work are important discussion points - doi: 10.1007/s00345-022-04038-8. Epub 2022 Jul 5. PMID: 35790535.: - doi: 10.1097/JU.0000000000002205. Epub 2021 Sep 22. PMID: 34547922.
Reply: We are very grateful for this objection and have incorporated all of your recommended studies in our discussion which has clearly helped to improve the understanding of the nerve sparing in patients undergoing radical prostatectomy for high risk disease.
As such, Whether or not neurovascular bundle preservation should be attempted in high risk prostate cancer patients undergoing radical prostatectomy remains a matter of debate. With this regards, continuous technological progressed has been associated with improved oncologic and functional outcomes. For instance, based on mpMRI bidimensional images, 3D- prostate models have been created to be used intraoperatively to allow for modulation of the nerve sparing. Checcucci et al could demonstrate limited occurrence of PSM, especially in patients with extracapsular extension at mpMRI or presence of pT3 prostate cancer on final histology, respectively.
On the other hand, Martini et al described a personalized approach to determine the grade of attempted nerve sparing by automated interaction detection. At this, a machine-learning partitioning algorithm was applied to identify risk groups by predicting extracapsular extension on final pathology, contralaterally to the prostate lobe with clinically high risk disease. They were able to show that wide bilateral excision in men with unilateral high risk disease is not justified and concluded that full and incremental nerve sparing in case of contralateral low and intermediate risk for extracapsular extension, respectively, is safe from an oncological stand point.
3. In addition, beyond nerve sparing other factors are critical to functional recovery such as the type of approach (for example retzius-sparing approach) and the reconstruction of supporting structures. In this regard I advise you to add in your article: - doi: 10.23736/S2724-6051.22.05179-5. PMID: 36629812. doi: 10.1111/bju.14716. Epub 2019 Mar 15. PMID: 30801887
Reply: We completely agree with the reviewer and have amended the discussion part accordingly. The fact that patients with unilateral nerve sparing may experience the same satisfactory functional outcomes compared to those with attempted bilateral neurovascular bundle preservation implicates that other factors critical to functional recovery have to be taken into account when analyzing differences in urinary continence and erectile function after radical prostatectomy. Surgical technique significantly changed by the adventages of robotics, with many different techniques and approaches (e.g. standard anterior prostatectomy vs posterior Retzius-sparing radical prostatectomy) that have been developed over the years to improve both functional and oncological outcomes. As such, the Retzius-sparing approach has been evaluated in various studies. The aim of this technique is to spare the anatomical structures surrounding the prostate by approaching the gland posteriorly and thus keeping the anterior connection between the pubis and the bladder intact. Galfano et al. showed in a large multicentric study that Retzius-sparing radical prostatectomy is feasible and safe also in the setting of high-risk prostate cancer. In terms of functional outcomes two recent systematic reviews of comparative studies reported earlier recovery of urinary continence when performing the Retzius-sparing approach as compared to the standard approach, even in the subpopulation of high-risk patients where the early continence is intrinsically harder to achieve due to a more disruptive surgery compared to low-risk disease. It is expected that future technological improvements and the creation of a standard training program may further increase the adoption of Retzius-sparing approach.
Furthermore, the reconstruction of supporting structures seem to positively impact functional outcomes as well. In this context, the total anatomical reconstruction technique during anterograde robotic radical prostatectomy, described by Manfredi et al, is a ‘‘tension-free’’ anastomosis technique that aimed to restore the anterior and posterior supports to the sphincter. With this technique authors could demonstrate excellent results in the early recovery of urinary continence.

Reviewer 3 Report

Comments and Suggestions for Authors

This is a retrospective study involving patients with high-risk prostate cancer, subject to radical prostatectomy with neuro-vascular bundle preservation.

1.     Authors should provide their institutional emails

2.     Keywords; should not be repeated from the title.

3.     The introduction section should be extended. The authors could provide more information for the reader to understand the manuscript topic.

4.     Selection criteria are poorly explored. The only criteria stated is: “All patients undergoing open radical prostatectomy for high risk prostate cancer were included.” If the prostatectomy is performed and for any reason nerve removal was necessary. After this phrase, the author says: “Nerve sparing was performed whenever possible,…”. Thus even without nerve sparing the patient was included? The authors could clarify this section.

5.     Figures show poor quality. Please, increase all image quality.  

Author Response

This is a retrospective study involving patients with high-risk prostate cancer, subject to radical prostatectomy with neuro-vascular bundle preservation.
1. Authors should provide their institutional emails
Reply: institutional emails of all authors have been provided on the title page of the manuscript.
2. Keywords; should not be repeated from the title.
Reply: We are grateful for this input. The key word ‘high risk prostate cancer’ has been replaced by multi-parametric magnetic resonance imaging,
3. The introduction section should be extended. The authors could provide more information for the reader to understand the manuscript topic.
Reply: The reviewer’s comment is greatly appreciated. We have extended the introduction accordingly and have taken the aspects to safely plan nerve sparing attempts more into account.
4. Selection criteria are poorly explored. The only criteria stated is: “All patients undergoing open radical prostatectomy for high risk prostate cancer were included.” If the prostatectomy is performed and for any reason nerve removal was necessary. After this phrase, the author says: “Nerve sparing was performed whenever possible,…”. Thus even without nerve sparing the patient was included? The authors could clarify this section.
Reply: We are grateful for this objection and agree that this section has not been explained clearly enough. We have created a new Table 1a ‘Inclusion criteria to attempt nerve sparing radical prostatectomy’. Table 1 has been renamed to Table 1b and 1c.
We have adapted the section to explain that in this study, all patients undergoing open radical prostatectomy with any degree of nerve sparing (no, unilateral or bilateral nerve sparing) for prostate cancer at our institution from 1996 to 2020 were included. In brief, nerve sparing was performed if there was no ipsilateral palpable induration on digital rectal examination, no ipsilateral capsular involvement in the preoperative MRI, and no contraindication intraoperatively. For inclusion criteria to attempt nerve sparing radical prostatectomy we then further referred to the new Table 1a.
5. Figures show poor quality. Please, increase all image quality.
Reply: We appreciate this remark. All figures have been revised and are provided in a PDF format.

Reviewer 4 Report

Comments and Suggestions for Authors

The article "Functional impact of neuro-vascular bundle preservation in high risk prostate cancer without compromising oncological outcomes: a propensity-matched analysis" presents an interesting topic that has been the subject of study in recent years.

In my opinion, the article has some revisions that need to be made to make it more enlightening:

- The tables and figures are quite confusing. Sometimes, the authors divide a and b, but the information is duplicated. For example, in Table 1 b, wouldn't it be easier to create two comparison columns and put them in Table 1 a and just call it 1?

- Table 2 appears after Figure 2, but Table 2 is described first in the text.

- Overall, the authors should better divide the results to clarify the article. For example:

- Compare results, of the whole cohort with ns vs unilateral vs bilateral

- Low-risk vs high-risk results;

- High-risk distributed results;

- Figure 2 has no color legend;

- Aren't the results regarding surgical outcomes a duplication of those seen in the published article? "Oncological outcomes after attempted nerve-sparing radical prostatectomy (NSRP) in patients with high-risk prostate cancer are comparable to standard non-NSRP: a longitudinal long-term propensity-matched single-centre study." The authors should focus on the innovative results of urinary and incontinence markers and cite the published article.

Comments on the Quality of English Language

The English of the article should be proofread by a qualified person for confusing sentences and errors.

Author Response

The article "Functional impact of neuro-vascular bundle preservation in high risk prostate cancer without compromising oncological outcomes: a propensity-matched analysis" presents an interesting topic that has been the subject of study in recent years.
In my opinion, the article has some revisions that need to be made to make it more enlightening:
1. The tables and figures are quite confusing. Sometimes, the authors divide a and b, but the information is duplicated. For example, in Table 1 b, wouldn't it be easier to create two comparison columns and put them in Table 1 a and just call it 1?
Reply: We appreciate this objection. We have renamed tables where reasonable. With regards to Table 1a and 1b (now Table 3a and 3b) we feel that combining them would be difficult given that Table ‘a’ includes patients stratified by the grade of nerve sparing whereas Table ‘b’ includes patients stratified by the risk profile. As such, we feel that combining all 6 subgroups may be more confusing.
2. Table 2 appears after Figure 2, but Table 2 is described first in the text.
Reply: We have renumbered Tables according to the chronological order in the manuscript. Tables and Figures are numbered independently, eg. Table 1 to Table 4, and Figure 1 to Figure 3. Therefore, Table 2 may appear after Figure 2.
3. Overall, the authors should better divide the results to clarify the article. For example:
Compare results, of the whole cohort with ns vs unilateral vs bilateral
- Low-risk vs high-risk results;
- High-risk distributed results;
Reply: We appreciate this objection and agree that the analysis comprises many parts which may make it difficult to follow. We have therefore analyzed functional outcomes as observed (no adjustment) after inverse probability of treatment weighing stratified by grade of nerve sparing and risk group (low- and intermediate risk vs high risk), see Table 5a. Likewise, oncological outcomes as observed (no adjustment) are reported as survival data after inverse probability of treatment weighing stratified by grade of nerve sparing and risk group (Table 5b).
As such, functional outcome 2 years after surgery showed similar patterns in low or intermediate risk and high risk patients when stratified by grade of NS. The same yields for oncological outcomes within 10 years. This has been added to the results section.
We have the impression that this tabularization of functional and oncological outcomes may increase understanding and overview of the results of the entire cohort and subgroups.
4. Figure 2 has no color legend
Reply: Thank you for this remark. Color legend to Figure 2 has been added.
5, Aren't the results regarding surgical outcomes a duplication of those seen in the published article? "Oncological outcomes after attempted nerve-sparing radical prostatectomy (NSRP) in patients with high-risk prostate cancer are comparable to standard non-NSRP: a longitudinal long-term propensity-matched single-centre study." The authors should focus on the innovative results of urinary and incontinence markers and cite the published article.
Reply: We appreciate this question. Whereas only high risk prostate cancer patients (n=726) have been included in the study ‘Oncological outcomes after attempted nerve-sparing radical prostatectomy (NSRP) in patients with high-risk prostate cancer are comparable to standard non-NSRP: a longitudinal long-term propensity-matched single-centre study’, we have included the consecutive series of all n=1340 patients undergoing radical prostatectomy for any prostate cancer (i.e., low, intermediate and high risk cancer). As such, the aim of this study was not just toinvestigate functional and oncological outcomes of high risk patients but of the entire prostate cancer cohort, and also compare those risk groups with regards to those outcome measures.
This has been specified in the introduction to say that comparison of the defined outcome measures between the two risk groups (low and intermediate vs high risk) is necessary to draw conclusions of the impact of high risk disease on functional and safety outcomes.
We have tried to bring the different and innovative results of functional and oncological outcomes, especially those with high risk disease into a more prominent perspective and cited the published article accordingly.
In the discussion section we specified that s significant benefit of unilateral or bilateral nerve sparing over no- nerve sparing was observed in both risk groups (low- and intermediate risk, and high risk prostate cancer) with respect to functional outcomes. We then further specified that in the sub-cohort of patients with low or intermediate risk, a higher proportion of patients reached erectile function recovery, corresponding to a lower proportion in the high risk subcohort. However, there was no substantial benefit of bilateral nerve sparing with respect to urinary continence outcomes in both risk groups.

Round 2

Reviewer 4 Report

Comments and Suggestions for Authors

The authors gave satisfactory answers to all the questions asked above.

Comments on the Quality of English Language

I recommend a final revision to correct small errors.